# Role of Epigenetic Factors in Response to Stress and Establishment of Somatic Memory of Stress Exposure in Plants

**DOI:** 10.3390/plants12213667

**Published:** 2023-10-24

**Authors:** Igor Kovalchuk

**Affiliations:** Department of Biological Sciences, University of Lethbridge, Lethbridge, AB T1K 3M4, Canada; igor.kovalchuk@uleth.ca

**Keywords:** stress tolerance, plants, DNA methylation, abiotic and biotic stresses, priming, epigenetics, stress memory

## Abstract

All species are well adapted to their environment. Stress causes a magnitude of biochemical and molecular responses in plants, leading to physiological or pathological changes. The response to various stresses is genetically predetermined, but is also controlled on the epigenetic level. Most plants are adapted to their environments through generations of exposure to all elements. Many plant species have the capacity to acclimate or adapt to certain stresses using the mechanism of priming. In most cases, priming is a somatic response allowing plants to deal with the same or similar stress more efficiently, with fewer resources diverted from growth and development. Priming likely relies on multiple mechanisms, but the differential expression of non-coding RNAs, changes in DNA methylation, histone modifications, and nucleosome repositioning play a crucial role. Specifically, we emphasize the role of BRM/CHR17, BRU1, FGT1, HFSA2, and H2A.Z proteins as positive regulators, and CAF-1, MOM1, DDM1, and SGS3 as potential negative regulators of somatic stress memory. In this review, we will discuss the role of epigenetic factors in response to stress, priming, and the somatic memory of stress exposures.

## 1. Introduction

The response to stress is genetically predetermined, since it has been honed by many years of evolution and ancestral environmental exposures. To be able to respond to the environment in a manner similar to their parental cells, each cell has to undergo faithful DNA replication [1]. In addition, all epigenetic marks need to be reproduced. In a stable environment, daughter cells look similar to parental cells in their genetic and epigenetic makeups and often in the set of various metabolites, with variations attributed typically to the developmental stage or tissue specificity (Box 1). Fluctuations from a stable environment trigger different responses from plants, occurring on biochemical, molecular, and cellular levels, and including changes in primary and secondary metabolites and epigenetic marks, allowing plants to survive stress [2,3]. When stress is no longer present, most of these changes disappear [3], but some are maintained [4], allowing daughter cells to receive the information about the response to stress, allowing plants to acquire partial protection (Box 1). This somatic stress memory could include various metabolites, as well as changes in DNA methylation or modifications of histone tails [4]. This was recently demonstrated in rice in response to salt—biochemical and epigenetic changes were noted, correlating with stress tolerance [4]. Similarly, the metanalysis of reports on the memory of stress applied prior and during seed germination showed that the memory was based on the changes at the level of chromatin reorganization, alternative transcript splicing, metabolite accumulation, and autophagy [5]. This memory allows the plant to perform better when stress returns. The most well-known examples of such a response to stress are acclimation and adaptation [6].

Box 1Changes in response to environment.**Adaptation**—the process of permanent changes in plant metabolism and
phenotype in response to a changed environment, observed in an entire
population of plants.**Acclimation**—the process of adjustment to changes in the environment,
allowing individual plants to maintain fitness and continue/complete their
developmental process. Typically, this is a short-term somatic process. Some
scientists consider the inheritance of this response as transgenerational
acclimation, also referred to as phenotypic plasticity. The changes are lost
when the stimulus is not maintained.**Normal environmental conditions**—fluctuations in abiotic growth conditions,
including temperature, water, light regimes, and biotic interactions to which
a specific plant species is adapted. Certain conditions may be the norm for
one species or population, but not for another.**Priming**—exposure to a mild stress that results in a higher
tolerance to a more severe stress of the same or similar nature. Priming
effect, sometime referred to as hardening, may be heritable in nature,
representing an intergenerational memory of stress.**Latent period**—time between the stress exposure and accumulation of
molecular and physiological changes required for stress tolerance.**Somatic stress memory**—the features of acclimation or tolerance (or
intolerance), be it phenotypic or epigenetic in nature, to subsequent stress
exposures.

In this review, the somatic stress response and priming will be described in more detail, and various mechanisms of epigenetic response to stress, including chromatin decondensation, the role of chromatin modifiers, histone modifications, changes in DNA methylation, the activity of non-coding RNAs, and RNA methylation, will be presented.

## 2. General Stress Response: Priming

An active metabolism, including photosynthesis, cellular respiration, and other physiological activities associated with the function of chloroplasts, peroxisomes, and lysosomes, poses a continuous challenge for any organism [7]. These internal stresses produce free radicals that are either damaging molecules such as DNA, RNA, protein, or lipid directly or trigger changes in these molecules by oxidation or via a variety of signaling pathways [8].

Avoidance is not part of a plant’s mechanism of stress response. Environmental stimuli represent external stresses that include, but are not limited to, changes in light intensity, temperature fluctuations, water and nutrient availability, wind, and other mechanical stimuli, as well as an entire realm of biotic interactions that include physical and chemical influences. Several good reviews cover the effects of different types of abiotic and biotic stresses on plants [9,10]. There is a plethora of changes in plant physiology, photosynthesis, respiration, water conductance, osmotic pressure, thermoregulation, and many others in response to stress that are well described in various reviews [11,12].

To survive these environmental stimuli, individual organisms have genetically programmed mechanisms of response. When stress persists, species also have to develop new adaptive changes in order to survive long term. Stress avoidance is a common response to stress in mobile organisms. Not being able to escape external stresses, plants are limited to mechanisms of tolerance and resistance. Adaptive metabolic changes in somatic cells [13] and heritable transgenerational changes are also part of advanced survival mechanisms. Through the process of evolution, organisms have developed various adaptive mechanisms of survival, and plants seem to be very efficient in doing that [14]. All the above-mentioned mechanisms of stress response require the rapid and efficient perception of stress and activation of signalling cascades, followed by immediate epigenetic changes, gene expression changes, changes in metabolism, and phenotypic changes, as well as short-term acclimation or long term epigenetic and phenotypic adaptation [15].

The mechanisms of general stress response also include acclimation or adaptation to stress, although to be effective, they require priming with milder stress [16]. The response to stress and the initiation of priming may be triggered by changes in plant hormonal levels; the levels of secondary messengers, such as Ca^2+^, free radicals, and phospholipids; and the activity of signal sensors and transducers, such protein kinases, transcription factors, and ubiquitination machinery, as well as the entire spectrum of epigenetic regulators [17,18,19,20]. Though these mechanisms are commonly functioning in somatic tissues during the growth of an organism, they may also have an intergenerational nature, and can be referred to as intergenerational acclimation [21].

The time and the level of molecular and physiological responses to stress vary for different stresses and depend on previous stress encounters [22]. Figure 1 shows several possible scenarios of priming, followed by the second stress exposure. The first scenario shows stress exposure not preceded by priming. In this case, the stress response and tolerance strictly depend on two parameters, the genetically predetermined response and the transgenerational memory of stress exposure, if such is present. The second scenario shows the priming event, followed by the second stress exposure. In this case, the latent period is likely shorter and the stress tolerance is higher and has a longer duration of time. The third scenario demonstrates priming, followed by a delayed second stress response. It can be hypothesized that stress tolerance would likely be similar to the scenario with no priming or likely somewhere between the “no priming” and “priming” scenarios. The next scenario is a “double stress” scenario; it can be hypothesized that if a similar stress occurs again in a reasonably short interval after the first stress, there may be even higher stress tolerance achieved. Finally, in a “double priming” scenario, it can be assumed that short encounters of mild stress occurring in a reasonably short period of time may result in elevated stress tolerance when compared to a single priming. Such “pulsed” priming appeared to work much better to build UV protection when compared to continuous priming [23].

The length of stress memory, and, thus, the effect of priming, likely depends on multiple parameters, including the time of priming, as well as the type, time, and severity of stress [24,25]. It is not clear whether the effect of priming diminishes in a linear or non-linear manner. It likely depends on the number of cell divisions between priming and stress application, since cells have a chance to maintain or reprogram established epigenetic marks, whether these are DNA or histone modifications or the accumulation of non-coding RNAs, during each cell division [26]. Indeed, it can be assumed that the more cell divisions the occur, the more chances these epigenetic marks can be “diluted” in the population of new cells if these marks are not maintained due to the absence of the triggering stimulus. It is also hypothesized that a gradual or incremental increase in stress may result in better stress tolerance upon future encounters compared to an abrupt exposure to a high level of stress [27]. One other proposed mechanism of “forgetting” the stress exposure is the autophagy—cells carrying the stress information are simply eliminated [28]. It is not clear whether such a mechanism is active and efficient in response to all stresses.

Since epigenetic changes may be of critical importance for the establishment of the response to stress and priming, the epigenetic changes occurring in stress-exposed plants will be covered in detail.

## 3. Epigenetic Regulation of Response to Stress

Epigenetic regulation is a critical component of the short- and long-term response to stress. Epigenetic regulation typically involves DNA methylation, histone modifications, and the activity of various non-coding RNAs [29]. Changes in nucleosome positioning, the redistribution of heterochromatin and euchromatin in the nucleus, and the differential binding of chromatin-modifying proteins (excluding histones) and methyl-CpG binding domain (MBD) proteins to DNA are also involved in the efficient response to developmental cues and environmental factors [30]. These epigenetic changes are passed from one somatic cell to another upon cell division, allowing for the effects of acclimation to occur and forming the so-called “transcriptional memory”, the epigenetic timestamp of the best possible transcriptional response to stress [31]. For example, the control of flowering upon transition from the vegetative to reproductive stage depends on the status of FLOWERING LOCUS C (FLC) acting as a molecular switch allowing the temperature signal received during vernalization to be remembered [32].

### 3.1. Changes in Chromatin Structure in Response to Stress

Chromatin is a complex structure consisting of DNA twice-wrapped around the octamer histone core, representing a nucleosome, together with the linker histone H1 and various chromatin-binding non-histone proteins. The chromatin structure is highly dynamic, representing regions of active transcription (euchromatin) or regions with poor transcriptional activity (heterochromatin), with the latter consisting of facultative and constitutive heterochromatin [33]. Various modifications of histone tails either result in the change of affinity of histones to DNA (histone acetylation) or a change in the tail structure, allowing the recruitment of a different kind of readers (histone methylation), thus regulating chromatin condensation and gene expression [34]. In addition, various non-histone proteins, including chromatin remodeling factors (CRF), also actively regulate chromatin condensation and, thus, processes such as transcription, replication, and DNA repair [35].

#### 3.1.1. Heterochromatin Decondensation and Activation of Transposable Elements in Response to Stress

The exposure to stress leads to the removal of nucleosomes from specific genomic locations, nucleosome repositioning [36], the loss of heterochromatin, and the activation of repetitive elements [37]. These events may or may not require changes in histone modifications. A study by Pecinka et al. (2010) showed that long-term exposure to heat in *Arabidopsis thaliana* resulted in the activation of some repetitive elements [38]. Surprisingly, the activation occurred without the loss of DNA methylation and with only minor changes to histone modifications. Repetitive elements were primarily activated by the loss of nucleosomes and heterochromatin decondensation. The recovery from stress was characterized by nucleosome loading and transcriptional silencing. Curiously, in Chromatin Assembly Factor 1 (CAF-1) mutants impaired in the chromatin assembly, the recovery stage and nucleosome loading were considerably delayed [38]. The substantial dissociation of heterochromatin was observed beyond the recovery phase when silencing and nucleosomes had been reinstalled; the loss of heterochromatin was observed in differentiated tissues of plants exposed to heat, and it lasted in the exposed leaves until they started to show signs of senescence. ***These experiments demonstrated that the stress memory can persist in the form of chromatin changes long after stress is removed. It also suggests that CAF-1 may act as a negative regulator of the establishment of priming/stress memory.***

A similar heterochromatin decondensation was observed in 2-day-old Arabidopsis plantlets in response to cell culturing, although regular chromocenters were formed in a stepwise process after a longer period in culture [39]. The loss of heterochromatin also occurred in older plants upon floral transition in development; however, the heterochromatin decondensation was not sufficient for repeat activation. Thus, local heterochromatization occurs during normal physiological and developmental processes and (un)specific responses to stress. Indeed, when plants were exposed to low-light stress, heterochromatin decondensation was more permanent and was directed toward areas with repetitive elements [39]. The important role of differential chromatin compaction for the response to stress was demonstrated in the work of Tessadori et al. (2009). The authors analyzed 21 accessions of Arabidopsis grown at different latitudes (and, therefore, different light intensities) and found a direct positive correlation between the latitude and the level of chromatin condensation; the higher the latitude, the higher the chromocenter compaction, and vice versa [39]. The authors suggested that such changes in chromatin compaction are part of the acclimation process in Arabidopsis.

So, is heterochromatin decondensation at genomic repeats a common response to stress? Pecinka et al. (2010) argue that this does not seem to be the case, as they did not observe this phenotype after freezing or UV-C irradiation [38]. Also, exposure to an abiotic stress may interfere with a plant’s capacity to withstand biotic stress. Indeed, even moderately increased temperatures can reduce biotic stress resistance by pathogens. In plants exposed to long-term heat stress, the activation of some repetitive elements is paralleled by silencing and the transcriptional repression of repetitive loci carrying clusters of resistance genes [38]. More recently, Pecinka’s lab reported the presence of heat-responsive elements (HRE) in several transposons, including *ONSEN*, *COPIA37*, *TERESTRA*, and *ROMANIAT5*, in *A. lyrata* and *A. thaliana*; HRE likely play an essential role in chromatin remodeling at transposable elements [40]. The response to heat may activate transposons in a methylation-dependent and independent manner. Recent work demonstrated that the exposure of the Arabidopsis H1 histone mutant to heat results in the activation of pericentromeric Gypsy elements without methylation loss, whereas non-pericentromeric Copia elements became activated in parallel with the loss of DNA methylation [41].

An additional level of complexity is achieved by differential heterochromatin decondensation in response to heat stress in different cells; the nuclei of meristematic cells do not undergo heat-induced decondensation [38]. This makes sense; if one considers that the heat stress response is transient in nature, it should largely occur in somatic tissues only; the lack of changes in the meristem indicates a safeguarding mechanism for minimizing epigenetic and, possibly, genetic changes in the germ line. This further supports the hypothesis that ***decondensation is a controlled process that occurs only either during specific stages of plant development or in response to specific stresses*** such as heat and high-light intensity stresses. Moreover, ***exposure to these stresses may result in the transcriptional activation of heterochromatin-embedded genes in differentiated cells, but not in dividing cells***.

#### 3.1.2. The Role of CRFs in Response to Stress

CRFs are ATPases from the sucrose nonfermenting 2 (Snf2) family that are involved in chromatin remodeling in eukaryotes. Their classification is beyond the scope of this review, and it has been covered well recently [35]. The role of CRFs in response to stress is to regulate gene expression, replication, and genome stability by interacting with DNA methylation, histone modification, and RNA processing machineries. In this review, only several important CRFs, including SWI2/SNF2 subfamily protein BRAHMA (BRM), Lsh2 subfamily protein DECREASED DNA METHYLATION1 (DDM1), and DRD1 subfamily protein DRD1, will be covered.

BRM is an ATPase essential for many developmental processes that are both chromatin-dependent (transcription regulation and maintenance of stress memory) and independent (such as pri-miRNA processing) [42]. BRM was shown to interact with many proteins involved in abiotic and biotic stress responses, including HEAT-STRESS-ASSOCIATED 32-kDa PROTEIN (HSA32) [36,43], histone H3-binding protein FORGETTER1 (FGT1), HD2C histone deacetylase, and H3K27me3 demethylase RELATIVE OF EARLY FLOWERING6 (REF6).

In view of the focus of this review, the main role of BRM in stress response is to maintain abiotic stress memory, with some specificity towards heat shock response. Specifically, FGT1 recruits BRM and CHROMATIN REMODELING 11 (CHR11)/CHROMATIN REMODELING 11 (CHR17) proteins to maintain low nucleosome occupancy at the heat shock memory genes in somatic tissues [36] (Figure 2). In addition, BRM activates the transcription of heat stress memory genes that remain upregulated in non-stressed offspring. This transgenerational activation involves the recruitment of BRM by REF6 near the transcription start site of heat shock memory genes, including BRM itself [44]. BRM has also recently been shown to have an overlapping role with the H2A variant H2A.Z, regulating the transcription of stress-responsive genes [45].

BRM also appears to play a role in response to drought and salt stresses. Exposure to these stresses lead to the accumulation of the phytohormone abscisic acid (ABA) [47]. Whereas under normal conditions, BRM represses the key regulator of ABA signaling pathway ABA-INSENSITIVE 5 (ABI5), in response to stress, a high level of ABA leads to the phosphorylation of BRM, resulting in its inactivation and the de-repression of ABI5 transcription [48,49]. There is also a link between energy sensor TARGET OF RAPAMYCIN (TOR) and BRM. TOR appears to repress genes associated with bistable chromatin state, driven by H3K4me3 and H3K27me3, and, thus, actively regulates transcriptional response to stress. BRM, together with CURLEY LEAF (CLF), which deposits H3K27me3 marks, activate TOR-repressed genes, allowing a rapid response to stress [50].

***Whereas BRM promotes the maintenance of heat shock-induced epigenetic memory***, ***DDM1 eliminates it*** [44,51]. This is evident from the longer persistence of the heat shock-induced transcriptional activation of heterochromatic loci in *ddm1* single mutant and in the *ddm1 mom1* double mutant when compared to the wild-type Arabidopsis. Of note, however, is the fact that heterochromatin hyperactivation status is transmitted to the progeny specifically in the *ddm1 mom1* double mutant, but not in the single *ddm1* mutant [51]. DDM1 also appeared to be involved in biotic stress response; together with another CRF member, SPLAYED (SYD), it represses the transcription of plant defense genes such as SUPPRESSOR OF *npr1-1* CONSTITUTIVE1 (SNC1) during a bacterial pathogen attack [52].

DDM1, together with MORPHEUS’ MOLECULE 1 (MOM1), modulates transcriptional gene silencing via DNA methylation [51,53]. The importance of *ddm1* for the control of DNA methylation is reflected by the fact that the *ddm1* mutant shows up to 70% reduction in global genome methylation, predominantly at heterochromatic regions [54]. Consequently, this triggers the activation of transposons and retrotransposons, the transcriptional activation of a previously silent disease-resistance gene array, and profound phenotypic instability amplified with every generation of self-propagation. The fact that ***ddm1-induced hypomethylation of various genes can be stably inherited through mitotic and meiotic cell divisions*** might be one of the reasons of the phenotypic instability [55].

Additional data on the DDM1 function were obtained from studies in maize [56]. ZmDDM1 is critical for CHG methylation, and less so for CG methylation in the heterochromatic regions, and is also required for the formation of mCHH islands. In addition, ZmDDM1 is required for the processing of 24-nt small interfering RNAs (siRNAs), suggesting it is likely involved in the RNA-direct DNA methylation (RdDM) pathway–gene expression regulation mechanism.

The mechanism of methylation loss in *ddm1* plants is not entirely clear, but it is possible that DDM1 regulates DNA methylation status via changes in histone methylation, the interaction with Arabidopsis MBD proteins (AtMBDs), or the regulation of RdDM pathway. It has been shown that *ddm1* exhibited a disrupted localization of AtMBDs at chromocenters, suggesting that DDM1 may facilitate the localization of MBDs at specific nuclear domains [57]. Recent papers have shed more light on the process. DDM1 appears to aid DNA methyltransferases in displacing the nucleosomes from the chromatin. It was recently found that DDM1 promotes the replacement of histone variant H3.3 by H3.1, which allows nucleosome displacement and DNA methylation to occur [58]. In *ddm1* mutants, the loss of the H3.3 chaperone HIRA allows partially restoring DNA methylation, further indicating the role of both DDM1 and H3.3 in the process [58].

One of the possible mechanisms of the involvement of DDM1 in the control of DNA methylation is the maintenance of CpG methylation at RdDM-targeted sequences after the RNA signal is removed. *ddm1* plants appear to be impaired in DNA repair and are sensitive to salt and methyl methane sulfonate (MMS) stress [59], likely due to the increased activity of transposons and retrotransposons and general dysregulation of heterochromatin. Also, previous research demonstrated that the propagation of *ddm1* mutants for nine generations resulted in a genome-wide redistribution of DNA methylation to euchromatin [60].

Another SWI/SNF-like protein, DRD1, represents a novel plant-specific chromatin-remodeling factor that is required for the RNA-directed de novo methylation of target promoters [61]. It is also necessary for the total loss of de novo DNA methylation after the RNA silencing trigger is withdrawn. DRD1 interacts with two other factors, NRPD1b and NRPD2a, which represent subunits of a novel, plant-specific RNA polymerase, pol IVb. DRD1 and the pol IVb complex act downstream of the ncRNA biogenesis pathway. They direct reversible silencing of euchromatic promoters in response to RNA signals possibly through the recruitment of DNA methyltransferases for the methylation of homologous DNA sequences. It is noteworthy that among putative DRD1 targets, there are DNA glycosylases, ROS1 and DME, which are involved in active DNA demethylation. The down-regulation of *ROS1* in *drd1* and *pol IVb* mutants confirms the importance of the DRD1/pol IVb pathway for the active loss of induced de novo DNA methylation [62]. As such, it is possible that DRD1 may play a critical role for the somatic stress memory.

#### 3.1.3. The Role of Chromatin-modifying Proteins Lacking ATPase Domain

Another potential chromatin remodeling factor playing a role in the regulation of silencing and the inheritance of heterochromatin hyperactivation states is the nuclear protein MOM1. Despite the fact that MOM1 is evolutionarily related to chromodomain helicase DNA binding protein3 (CHD3), a chromatin remodeling protein, MOM1 does not contain the functional ATPase/helicase domain [63].

It is involved in DNA methylation- and histone methylation-independent silencing of repetitive sequences in *Arabidopsis* by preventing the transcription of 180-bp satellite repeats and *106B* dispersed repeats [64]. This suggests ***the existence of two distinct epigenetic silencing pathways: one that is DNA-methylation-dependent and another one that is DNA-methylation-independent***. Although MOM1 is involved in chromatin remodeling, the mutant is not hypersensitive to the DNA damaging agent MMS. Other chromatin modifiers, such as BRUSHY1 (BRU1), FASCIATA (FAS1), FAS2, and Replication protein A 2 (RPA2), are also dispensable for DNA methylation, but all of them are hypersensitive to the MMS-induced DNA damage.

***One of the first proteins implicated in the appearance of heat shock memory is the HEAT SHOCK TRANSCRIPTION FACTOR A2 (HSFA2)*** [65]. Besides its role in inducing the expression of genes that are on for 2-3 days in response to heat, HSFA2 also establishes the hyper-methylation of histone H3K4 at the loci associated with heat shock memory formation [66]. The *FGT1* gene provides another link between heat stress memory and the organization of the chromatin; FGT1 protein is essential for the sustained induction of *HSA32* and several other memory genes after heat stress [36]. FGT1 interacts with several CRFs and maintains low nucleosome occupancy throughout the heat stress memory phase [36]. Another link between the activation of heat shock proteins and heat stress memory appears to be the glucose and the energy sensor TOR-E2Fa signalling module [67]. In response to heat, glucose can promote chromatin acetylation of H3K4 trimethylation. Glucose-primed accumulation of H3K4me3 at heat stress memory-associated loci is mediated through HLP1 and requires the activity of TOR [67].

Other reports also indicated the link between chromatin maintenance and stress response. Mutants of a nuclear protein BRU1, encoded by the BRUSHY1 (BRU1)/TONSOKU (TSK)/MGOUN3 (MGO3) gene, involved in the maintenance of chromatin structure and epigenetic inheritance of chromatin states, were highly sensitive to genotoxic stress and were characterized by an increased frequency of intrachromosomal homologous recombination [68]. Recent report suggests that in Arabidopsis, ***in response to heat stress***, ***BRU1 is required to maintain a sustained induction of heat shock-induced memory-associated genes*** [69]. Curiously, BRU1 was dispensable for the increase in thermotolerance. Based on the data obtained using *bru1* mutants, the authors proposed a model where ***BRU1 mediates the faithful inheritance of chromatin states after DNA replication and cell division***.

BRU1 mutants are very similar to Chromatin assembly factor (CAF-1) component mutants, namely FASCIATA1 (FAS1) and FAS2, in various responses to stress; although, a report by Brzezinka et al. (2019) suggests that in contrast to BRU1, CAF-1 is not required for heat stress memory [69]. CAF-1 is known to be involved in priming plant defences via the modulation of nucleosome occupancy [70], whereas no such information exists for BRU1. These findings suggest that although CAF-1 and BRU1 likely function in connected molecular pathways, their functions only partially overlap.

### 3.2. The Role of Histone Modifications in the Response to Stress

Histone modifications play an important role in the response to stress and in the establishment and propagation of somatic stress memory [71,72,73,74,75].

In plants, transcriptionally active chromatin exhibits an enhancement of H3 and H4 acetylation and the trimethylation of lysine 4 from histone H3 (H3K4me3), whereas silent chromatin contains hypoacetylated H3 and H4, methylated lysine 27 (H3K27), and lysine 9 of histone H3 (H3K9) [76]. Histone acetyltransferases (HATs) and histone deacetylases (HDACs) modulate the expression of developmental and stress-sensitive genes. Multiple histone deacetylases are repressed by stress. Histone deacetylases associate with PWR (POWERDRESS) and HOS15 (HIGH EXPRESSION OF OSMOTICALLY RESPONSIVE GENES) proteins that are directly repressing stress response genes by modulating histone acetylation at H3K27, H3K36, H3K56, H3.3K27, and other sites [77].

There are many reports showing histone modifications in response to various stresses, so only some, including temperature, salinity, drought, and pathogen stresses, will be covered. For example, the deposition of active histone marks H3K9Ac and H3K4me3 on heat shock protein-encoding genes *HSP18*, *HSP22.0*, *APX2*, and *HSP70* occurs in response to heat stress. Histone H3K4 methyltransferases SET DOMAIN PROTEIN 25 (SDG25) and ARABIDOPSIS HOMOLOG OF TRITHORAX 1 (ATX1) increase H3K4me3 and decrease DNA methylation at stress response genes during heat shock stress recovery [78] (Table 1). GCN5 acetyltransferase is required for the response to heat stress, as the *gcn5* mutant is deficient in the induction of heat-responsive genes [79]. Exposure to heat, cold, and drought resulted in an increase in H3K9K14ac, H3K4Me2, and H3K4Me3 enrichment at several genes, including *FRK1*, *WRKY53*, and *NHL10*, involved in the immune response, which actually led to pathogen resistance [80] (Table 1). Curiously, mutants impaired in histone acetyltransferase1-1 (*hac1-1*) did not show this enrichment and were as sensitive to bacteria as plants that were not exposed to stress; so, no priming of pathogen resistance was observed [80].

Histone acetylation is enriched in the bodies of a number of cold-responsive genes, including COR15A and COR47 [81,82]. Also, ADA2b, an interacting partner of GCN5 acetyltransferase, is stimulated by cold-responsive transcription factor CBF1, resulting in H3 hyperacetylation at COR genes [83]. The role of deacetylases is not entirely clear, as they are implemented in regulating the cold stress response and acclimation in a positive and a negative manner. During cold exposure, CULLIN4-based ubiquitin E3 ligase complex degrades HD2C deacetylase and its partner HOS15, resulting in the increased level of histone acetylation at COR genes, leading to their expression [84]. It should be noted, however, that when cold stress is removed, histone acetylation marks are removed within 24 h of returning to normal conditions [85]. At the same time, HD2C deacetylase in Arabidopsis was also shown to activate COR genes upon cold stress [82]. HDA6 may also be required for cold acclimation, since an *hda6* mutant exposed to cold does not exhibit the same level of freezing tolerance as cold-treated wild-type plants [85].

Exposure to cold decreases the deposition of H3K27me3 in the promoters of COR15A gene and stress-responsive gene ATGOLS3 [86] (Table 1). The chromatin state with low H3K27me3 occupancy at COR15A and ATGOLS3 genes is inherited somatically, for up to three days [87], but does not result in a higher expression of these genes upon a second exposure to cold, suggesting the involvement of some other mechanisms [74].

**Table 1 plants-12-03667-t001:** Histone modifications in response to stress.

Stress/Histone Modification	Acetylation	Methylation	Phosphorylation/ Ubiquitination	Histone Variants	Reference
**Heat**					
H3K4 methyltransferases SDG25 and ATX1		↑ H3K4me3			[78]
HATs and H3K4 methyltransferases	↑ H3K9K14ac at *WRKY53*, *FRK1*, *NHL10*	↑ H3K4me2/3 at *WRKY53*, *FRK1*, *NHL10*			[80]
HATs like GCN5, HD2C	↑ H3K9ac at HSP18, HSP22, APX2, HSP70				[78,79]
FGT2 encodes a TYPE-2C PROTEIN PHOSPHATASE (PP2C); mutant is impaired in heat shock memory			??		[88]
H2A.Z—temperature sensor				↓ H2A.Z at HSP70	[89]
H2AX—phosphorylation ↑ after DNA damage				↑ γH2AX at damage sites	[90]
**Cold**					
Histone acetyltransferases (HATs) like GCN5,	↑ H3K9ac at COR15A, COR47				[81,82,83]
Histone deacetylases (HDACs), HD2C, HDA6	COR genes				[82,84,85]
HATs and H3K4 methyltransferases	↑ H3K9K14ac at *WRKY53*, *FRK1*, *NHL10*	↑ H3K4me2/3 at *WRKY53*, *FRK1*, *NHL10*			[80]
Demethylases		↓ H3K27me3 at COR15A, ATGOLS3			[86]
**Salt**					
HATs (GCN5?), HMT, demethylases	↑ H3K9/K14ac at *GCN5* and *CTL*	↑ H3K4me3 ↓ H3K9me2 ↓ H3K27me3			[91,92,93]
HATs	↑ H3K9/K27 at *POX* gene				[94]
HDAC inhibition	↑ H4ac at *AtSOS1*				[95]
Demethylases		↓ H3K27me3 at *HKT1* gene			[30]
Kinases			↑ H3Ser-10ph		[92]
**Drought**					
HATs, methyltransferases	↑ H3K9/K14ac at *NCED3*, *GOLS2*, *RD20*, *RD29A*, *RD29B*, *RD22*, and *RAP2.4*	↑ H3K4me3 at *NCED3*, *GOLS2*, *RD20*, *RD29A*, *RD29B*, *RD22*, and *RAP2.4*			[96,97]
HATs and H3K4 methyltransferases	↑ H3K9K14ac at *WRKY53*, *FRK1*, *NHL10*	↑ H3K4me2/3 at *WRKY53*, *FRK1*, *NHL10*			[80]
↓ methyltransferase CAU1		↓ H4R3sme2 at *ANACo55*			[98]
↓ demethylase JMJ17		↑ H3K4me2/3 at *OST1*			[99]
**Pathogen response**					
*AtHDAC19* induced by *Alternaria brassicicola* by JA	↓ H3K9ac at WRKY38 and WRKY62				[100,101]
*AtHDAC6* induced by jasmonic acid	??				[102]
*AtHDAC19* and *HOS15* negatively regulate immunity	histone deacetylation at NLR genes				[103]
TaHDT701 deacetylase makes wheat sensitive to *Blumeria graminis*	Defence response genes				[104]
Ubiquitination—resistance to many stresses			H2B monoubiquitination		[105,106]

HMT, histone methyltransferase; HAT, histone acetyltransferase; HDAC, histone deacetylase; JA, jasmonic acid; ↑ indicates upregulation; ↓ indicates downregulation; ??, unknown.

Salt stress also triggers the accumulation of H3K9/K14Ac and H3K4me3 and reduction in H3K9me2 and H3K27me3 repressive marks on salt stress-responsive genes [91,92]. H3K9/K14 acetylation in Arabidopsis in response to salt activates cell wall biosynthesis gene acetyltransferase *GCN5* and stress response gene chitinase-like (*CTL*) [93]. The body of the *Arabidopsis HKT1* gene, encoding the Na^+^ transporter, is normally highly enriched in H3K27me3, and salt exposure results in the removal of H3K27me3 from the *HKT1* body and the activation of its expression [30].

The accumulation of permissive chromatin marks H3K4me3 and H3K9Ac in the promoter or gene body of many drought responsive genes, including *NCED3*, *GOLS2*, *RD20*, *RD29A*, *RD29B*, *RD22*, and *RAP2.4*, results in an increase in the expression of these genes [96,97]; an abundance of H3K4me3 and H3K9Ac positively correlates with gene expression and drought tolerance [107]. Also, the severity of drought exposure positively correlates with an abundance of these histone marks [108]. In Arabidopsis, the expression of proline in response to transcription factor ANAC055 is suppressed by the activity of CAU1 histone methyltransferase, which establishes H4R3me2 at the promoter. In response to drought, CAU1 is downregulated, leading to a decrease in H4R3me2, the activation of *ANAC055* expression, and the accumulation of proline [98]. JMJ17 demethylase is a negative regulator of response to drought stress; JMJ15 binds to the chromatin of the *OPEN STOMATA 1* (*OST1*) gene, removing H3K4me1/2/3 marks and suppressing *OST1* mRNA production [99]. *JMJ17* mutants have high levels of H3K4me2/3, a higher expression of *OST1*, and are more drought tolerant.

The role of histone modifications is also well documented for pathogen response. HDACs, for example, have been implicated in defense against pathogens. The HC-toxin from *Cochiobolus carbonum* specifically targets HDAC activity, causing histone hyperacetylation in susceptible corn cultivars [109]. In Arabidopsis, the *AtHDAC19* gene is induced in a similar manner by the fungus *Alternaria brassicicola* or by the exogenous application of jasmonic acid (JA) [100]. The overexpression of the *AtHDAC19* gene enhances fungal resistance through the apparent activation of the Ethylene Responsive Factor 1 (ERF1), whereas silencing of the gene increases fungal susceptibility. The expression of another HDAC, AtHDAC6, was also shown to be induced by JA application [102]. *AtHDAC19* is possibly involved in Arabidopsis resistance to a bacterial pathogen, *Pseudomonas syringae*. The proposed mechanism may involve a decrease in histone acetylation through interactions of HDAC19 with WRKY38 and WRKY62, two transcription factors that repress the SA pathway [101]. The locus-specific suppression of transcription of these two WRKY genes results in the activation of the SA-dependent pathway and, thus, in the resistance to bacterial pathogens.

Histone deacetylases can also act as negative regulators of response to pathogens: TaHDT701 was shown to be a negative regulator of wheat defence responses to *Blumeria graminis* f.sp. *tritici* [104]; HDA9 negatively regulates the Arabidopsis response to pathogens by histone deacetylation at NLR genes [103].

Changes in histone phosphorylation, sumoylation, and ubiquitination in response to stress are less well documented. It was shown that H3S10ph accumulates in response to salt and cold stresses and the H3T3ph mark accumulates in pericentromeric regions in response to drought [110]. The monoubiquitination of H2B results in the activation of stress-associated genes in response to many abiotic and biotic factors [105,106]. FGT2 encodes a TYPE-2C PROTEIN PHOSPHATASE (PP2C) and can potentially dephosphorylate histones. Plants mutated in FGT2 have normal thermotolerance, but are impaired in the establishment of heat shock memory [88].

Besides various modifications of core histones, histone variants such as H2A.Z also play role in the response to temperature stress in Arabidopsis [111]. At moderately high temperatures, the tight wrapping of H2A.Z and the amount of H2A.Z are reduced at the promoter of heat-responsive genes such as HSP70. Changes in the expression of H1 linker histone variants are essential to an adaptive response to complex abiotic stresses, such as low light and ABA treatment [89]. H2A.Z was suggested to be a plant temperature sensor, as mutants unable to incorporate H2A.Z into nucleosomes have a constant thermomorphogenic response, that is, abnormal growth in response to prolonged exposure to ambient temperature [110]. Stresses that cause DNA damage induce the phosphorylation of another histone variant, H2AX, which plays an essential role for activation of strand break repair mechanisms [90].

More examples of the role of histone modifications in stress response are well covered in recent reviews [30,31,112].

### 3.3. The Role of DNA Methylation in Response to Stress

DNA methylation is the most versatile mechanism involved in the regulation of gene expression, including the inheritance of specific gene expression patterns through somatic or meiotic cell divisions. The control of DNA methylation in plants is complex, with symmetrical CpG and CpHpG and non-symmetrical CpHpH methylation established and maintained through multiple, partially redundant mechanisms (Box 2). De novo symmetrical methylation is established by the DOMAINS REARRANGED METHYLTRANSFERASE 2 (DRM2) with the help of ncRNAs of the RdDM pathway, while maintained by the METHYLTRANSFERASE 1 (MET1) in the CpG context and CMT2/CMT3 proteins in CpHpG context [113]. CMT3 is recruited to the repressive histone mark H3K9me2 [114], and, in turn, CMT3 binding to DNA can facilitate the recruitment of H3K9me2 [114]. In contrast, CpHpH methylation is established by DRM2 and maintained by DRM2 at short transposons in euchromatic regions and by CHROMOMETHYLASE 2 (CMT2) at large transposons in heterochromatic regions [115]. DRM2 uses 24-nt siRNAs to guide DNA methylation at euchromatic TEs [116,117], whereas DDM1 mediates the recruitment of CMT2 to pericentromeric H3K9me2 regions [118].

Stress may result in both hypo- and hypermethylation at specific genomic loci, and these changes may represent either a short-term change or a long-term strategy of response to stress [119]. Promoters of stress-responsive genes are often found to be hypomethylated [120,121], whereas methylation at other genomic loci may not be altered and, sometimes, may even be increased [121].

Box 2DNA methylation in plants.**CpG methylation**—symmetrical cytosine methylation, abbreviated
as CG or CpG; often found in clusters, especially in the promoter regions of
genes expressed in tissue-specific manner; it is established de novo with the
function of RdDM machinery and maintained with MET1 methyltransferase.**CpHpG**—symmetrical cytosine methylation, abbreviated as CpG or
CpHpG, where H is C, A, or T; it is established de novo with the function of
RdDM machinery and maintained with CMT2/CMT3 methyltrasferase.**CpHpH**—asymmetrical or nonsymmetrical cytosine methylation,
abbreviated as CHH or CpHpH, where H is C, A, or T; it is established de novo
through RdDM machinery with the function of DRM2; it is likely maintained by
DRM2 or CMT2, with the assistance of DDM1, depending on genomic location.**De novo methylation**—methylation of previously
unmethylated cytosines.**Maintenance methylation**—methylation of hemimethylated DNA, or
re-methylation of cytosines at CpHpH context, arising either after DNA
replication or DNA resynthesis during repair.**Demethylation**—passive loss of methylation due to the lack of
maintenance (low activity or eviction of the methyltransferase) or active
loss through the activity of base excision repair and DNA glycosylases.

Changes in DNA methylation in response to stress may occur due to many different mechanisms, including the activity of DNA methyltransferases; DNA demethylases such as ROS1, DME1, DML2, and DML3; a passive loss of methylation via the exclusion of DNA methyltransferases from the nucleus; changes in the activity of chromatin remodelling factors and effector proteins; and many other changes in proteins regulating the chromatin structure [113].

The importance of DNA methylation for the maintenance of gene expression patterns and genome stability is reflected by the fact that plants evolved a specific enzyme to excise methylated cytosines from DNA. ROS1 is a methylated cytosine-specific glycosylase that excises methylated cytosines through the process of base excision repair [122]. This enzyme is rather unique in plants, since it combines the function of a DNA repair enzyme with that of an active demethylating process. Curiously, in the *ros1* mutant, the expression of several transposons was found to be decreased due to an increase in methylation levels at CpHpG and CpHpH sites [123]. Active ***DNA demethylation is, thus, important in pruning methylation marks in the genome***, and even previously silent transposons need dynamic control by methylation and demethylation. Such control is required for the plant epigenome to efficiently respond to developmental and environmental cues. For more detailed information on the types of active demethylation processes, see the review by Li et al. [122].

Many stresses may directly influence the level of methylation in the genome. According to the literature, the salts of Cd, Ni, and Cr cause oxidative damage that induces DNA hypomethylation [124]. In addition to the activation of ROS1 mechanisms, the other potential hypomethylation mechanisms are: the activation of DNA damage-specific endonucleases, such as those associated with the formation of single-stranded breaks; the formation of premutagenic 8-oxo-2′-deoxyguanosine adducts in response to heavy metal stress, which strongly inhibit the methylation of adjacent cytosines; an increase in nicotinamide levels in response to oxidative stress—through its metabolite trigonelline, nicotinamide can trigger hypomethylation in the genome [125].

The immediate stress response of plant somatic tissues results in changes in methylation of various areas of the genome, with genes involved in stress response being primarily hypomethylated [126]. Under salt stress, the stress-responsive genes, including those involved in the JA pathway, undergo rapid hypomethylation, leading to increased jasmonoyl isoleucine (JA-Ile) content and JA signaling to confer salt tolerance. Also, exposure to cold causes demethylation and transcriptional activation at many different loci in different species, including transcription factors, stress-response genes, and transposons [123,127]. Likewise, massive demethylation changes were observed in Arabidopsis plants exposed to heat stress [128]. Demethylation is possibly associated with the lower activity of RdDM machinery, including the downregulation of RNA-DIRECTED DNA METHYLATION 4 (RDM4) [129]. In turn, RDM4 regulates the expression of transcription factors responsive to cold stress [31].

Though DNA hypomethylation at stress-responsive loci is often associated with stress response and stress tolerance, an increase in DNA methylation is also frequently observed. For example, salt stress triggered hypermethylation more commonly than hypomethylation in *Beta vulgaris* ssp. *maritima*, especially in the promoter-located CpG sites [121]. In Arabidopsis, the hypermethylation of a putative small RNA target region upstream of the *HIGH-AFFINITY K+ CHANNEL 1* (*HKT1*) gene, which encodes a transporter that mediates Na^+^ influx in plants coordinating salt tolerance, leads to salt sensitivity [130]. Also, a genotype- and tissue-specific methylation increase occurred in response to salt in wheat, downregulating the expression of wheat *HKT* homologs *TaHKT2;1* and *TaHKT2;3*, leading to a genotype-specific response to salt [131].

Though changes in DNA methylation may be important for priming and acclimation processes, there are reports indicating a stochastic nature of DNA methylation in response to stress. Arabidosis plants exposed to a recurring excess-light stress showed negligible stress-associated changes in DNA methylation; although, they demonstrated clear evidence of photoacclimation through enhanced photosystem II performance in exposed tissues [132]. This indicates that acclimation can occur without meaningful changes in DNA methylation.

### 3.4. Role of siRNAs and RdDM in Stress Response

Many types of ncRNAs have been implemented in the direct or indirect regulation of various mechanisms of stress response and acclimation [133]. In this review, only the aspects of the involvement of siRNAs in RdDM will be covered, as this mechanism is the most likely to be involved in the establishment of somatic memory of stress exposures.

RdDM is a versatile mechanism implemented in development and reproduction, such as the control of flowering, TE silencing, regulation of genome stability, cell-to-cell and systemic signalling, and response to stress [134]. RdDM resembles RNA interference mechanisms and involves small non-coding RNAs, Dicers, Argonautes, and RNA-dependent RNA polymerases. siRNAs involved in RdDM are commonly produced from transposable elements, viral RNAs, transgenes, and various endogenous transcripts. RdDM occurs through a concerted function of sequence-specific siRNAs, PolIV, DCL3, RDR6, DRM2, RDM4, and several other proteins. Two polymerases, Pol IV and Pol V, are involved in the RdDM pathway. Pol IV generates 26-50 nt Pol IV-dependent RNA precursors (P4-RNAs) that are then converted into double-stranded RNA (dsRNA) by RDR2, which are then processed by DICER-LIKE protein 3 (DCL3) into 24-nt siRNAs or in the absence of DCL3, by DCL1, DCL2, and DCL4 into shorter 21- or 22-nt siRNAs [135]. Together with AGO4 or AGO6, siRNAs are loaded into RNA-induced transcriptional silencing (RITS), where they pair with the nascent Pol V transcripts. RITS then recruits DRM2 to catalyze CHH methylation [135]. More details on de novo RdDM silencing and self-reinforcing loops can be found in Bond and Baulcombe [136] or He et al. [135].

RdDM plays an essential role in the suppression of transposon activity in response to stress. For example, transposon ONSEN, while activated by heat stress, can only transpose if RdDM components are intact [137]. Moreover, the heat tolerance is decreased in several RdDM mutants; plants deficient in NRPD2, the subunit of RNA polymerases IV and V, are more sensitive to heat [138].

Some direct and indirect evidence exists demonstrating the role of RdDM in somatic stress memory and the formation of epialleles. The restoration of the function of NRPD1, a polIV subunit, results in the formation of two sets of loci, those that are remethylated after restoration, and those that are not [139]. The latter ones contain higher levels of the H3K4me3 euchromatic mark, which interferes with the recruitment of the RdDM machinery, and the H3K18ac mark, which attracts ROS1 to antagonize RdDM. Those loci that can be remethylated lack H3K4me3 and H3K18ac; CG/CHG methylation at these loci serves as a memory mark that is targeted by RdDM, aiding in the formation of stable epialleles.

Prolonged heat exposure reactivates repetitive elements silenced by posttranscriptional gene silencing (PTGS). The transcriptional activation of these loci increases the level of double-stranded RNA (dsRNA), which are one of the sources of siRNAs. The formation of dsRNA requires SUPPRESSOR OF GENE SILENCING 3 (SGS3) [140]. SGS3 activity is, in turn, regulated by temperature—a shift from 22 °C to 30 °C results in a decrease in SGS3 protein levels and, thus, in a decrease in the abundance of many trans-acting siRNAs (tasiRNAs) produced from dsRNAs. Induced by heat stress, HSFA2 transcriptionally activates the H3K27me3 demethylase RELATIVE OF EARLY FLOWERING 6 (REF6), which, in turn, further derepresses HSFA2, establishing a somatically heritable feedback loop; they activate an E3 ubiquitin ligase, SGS3-INTERACTING PROTEIN 1 (SGIP1), ***which leads to SGS3 degradation***, ***the inhibition of tasiRNA biogenesis, and the formation and maintenance of somatic and transgenerational memory of heat stress*** [44].

RdDM was also shown to be essential for the regulation of the AtMYB74 transcription factor in response to salt stress [141]. The AtMYB74 promoter is heavily methylated and is targeted by several 24 nt siRNAs; in response to salt stress, the production of these siRNAs is strongly decreased, decreasing promoter methylation and derepressing AtMYB74 [141].

As a part of RdDM, heterochromatic siRNAs (hc-siRNA) promote the hyper-methylation of *CYTOKININ-OXIDASE 2.1* (*HvCKX2.1*) promoter in response to terminal drought stress in barley, accelerating shoot emergence in the progeny [142].

RdDM components were implicated in the regulation of long-lasting memory of the immunity to herbivory; Arabidopsis plants exposed to JA were more resistant to insect grazing and were more sensitive to pathogens [143]. This effect required MYC2/3/4 transcription factors and RdDM activity, as well as the DNA demethylase ROS1 and the small RNA (sRNA)-binding protein AGO1. RdDM and ROS1 activated the ATREP2 transposable element, producing 21 nt sRNAs that are associated with AGO1. The authors hypothesized that since ATREP2 TEs are enriched with sequences from induced resistance-related defence genes, sRNAs derived from these transposons establish or maintain JA-induced herbivore immunity [143].

### 3.5. Potential Role of Methylated RNA in Stress Response

RNA methylation is a common process, well-described in animals, but still poorly understood in plants [144]. Methylated adenine is most common (m^6^A), but cytosines (m^3^C) can also be methylated. Plants encode all classical writers, erasers, and readers of methylation. The importance of these proteins is shown by various phenotypic abnormalities, including altered growth [145,146] and the response to pathogens [147].

Thousands of genes were reported to be affected by RNA methylation [148]; this is especially common in organelles. Almost all chloroplast transcripts and 86–90% mitochondrial transcripts are methylated, suggesting a critical role of this process in the energy metabolism [149]. The methylation of ribosomal RNA in chloroplasts is crucial for chloroplast biogenesis, photosynthesis, and for a proper abscisic acid response [150].

The methylation of mRNA may alter its mobility. m^5^C-modified *TCTP1* (*TRANSLATIONALLY CONTROLLED TUMOR PROTEIN 1*) was shown to be actively transported to roots and change root growth [151]. Methylated mRNAs can move long distances and, thus, could serve as developmental and stress response signals. In Arabidopsis, the mutation of m^6^A writers resulted in a sensitivity to salt stress that correlated with the level of m^6^A [152]. One of the writers, VIR protein, modulated the balance of reactive oxygen species by destabilizing the mRNA negative regulators of salt stress such as ATAF1, GI, and GSTU17 [152]. Similarly, the exposure of barley roots to cadmium stress resulted in significant changes in m^6^A levels in over 400 genes, including Cd-responding regulatory genes of MAPK, WRKY, and MYB groups [153].

Although there are not many reports indicating the direct role of methylated RNA in response to stress and stress memory, m^6^A and m^3^C in stress-responsive mRNAs and perhaps miRNAs [154] have to be considered as potential candidates.

## 4. Summary of Inducers and Blockers/Erasers of Priming/Stress Memory

Stress in plants is sensed through various mechanisms, including HSP sensing misfolded proteins, phyB sensing changes in light and temperature, Open Stomata1 (OST1) protein kinase sensing cold, changes in membrane fluidity, Ca^2+^ influx, hormone concentration, and many more [155]. Eventually, these signals are integrated into the response through signal transduction cascades and transcription factors, leading to the activation of various stress responsive genes (Figure 2). There are many stress-responsive genes that are either specific to a certain stress or to a broad range of stresses, and their main function is to activate various mechanisms of immediate response, including massive epigenetic changes to deal with the stress. These mechanisms in parallel trigger many other processes, including the release of PTGS from repetitive elements and the maintenance of reprogramming at the stress-responsive gene loci.

As mentioned in the section describing the role of RdDM, the release of PTGS results in the production of the RdDM fuel, dsRNAs, leading to the accumulation of various siRNAs. This process is regulated on multiple levels, including all components of RdDM mentioned earlier, but is also regulated by stresses like heat, which leads to the inactivation of SGS3, a protein positively regulating siRNA production (Figure 2). As a result, the level of siRNAs that are able to target TEs and loci with various stress-responsive genes depletes, allowing the maintenance of an open chromatin status around both. At the same time, stress-induced transcription factors activate a variety of stress-response genes and lead to local and stress-specific changes in histone modifications and nucleosome occupancy (Figure 2). One of the central roles in the establishment and maintenance of low nucleosome occupancy is played by BRM; BRM, together with CHR17, is recruited to stress-responsive genes by FGT1 and H3-binding protein [36] (Figure 2).

A parallel or alternative pathway is through REF6, which is also able to bind and activate BRM. BRM can interact with HFSA2 and promote H3K4me3 and the expression of the stress-responsive gene; this activity is also maintained by the deposition of the H2 variant H2A.Z histone (Figure 2).

Other chromatin modifiers, like BRU1, CAF-1, MOM1, and DDM1, are also implemented in various aspects of stress memory; whereas BRU1 is a positive regulator, CAF-1 appears to be a negative regular. Mozgova et al. write: “The mechanism limiting CAF-1-independent nucleosome assembly at promoters of many defense-related genes is important for plant fitness as it allows establishment and maintenance of a primed gene expression state that contributes to an efficient defense against pathogens” [70]. DDM1 and MOM1 may also interfere with the establishment of stress memory or be involved in its eraser, as plants mutated in these two proteins are enhanced in heat stress memory [51]. DDM1 likely prevents the inheritance of most stress-induced changes in histone methylation and associated changes in the methylation at the stress-responsive genes; that is why the hypomethylation of these genes is often inherited mitotically and meiotically in the *ddm1* plants [55] (Table 2).

Finally, proteins like ROS1 are involved in the eraser of various methylation patterns, and counteract RdDM1 activity and, thus, can be implemented in both the establishment and erasure of the stress memory, depending on the stress and the locus in question. Similarly, DNA methylation in general and histone modifications may play dual roles; although, it is more likely that local regions of permissive chromatin are more important for the establishment of stress memory (Table 2).

What are other potential mechanisms of maintaining the memory of stress exposure? It was proposed that alternative splicing is a “molecular thermometer” in plants, allowing the regulation of plant development [156]. H3K36me3 is involved in the regulation of temperature-induced alternative splicing [157], as it promotes the opening of the chromatin, allowing for a fast rate of elongation during transcription, benefitting splice site skipping. Thus, variations in temperature could be memorized by deposition H3K36me3 at the specific chromatin regions, leading to a specific splicing pattern, allowing a response to specific temperature [157].

## 5. Concluding Remarks

Plants, as any other species, need to balance their resource allocation. This is especially important upon stress exposure. The mechanisms of priming and hardening have evolved to give plants an edge in their survival in stressful environments.

In this review, I summarized how the stress response is regulated on an epigenetic level and introduced priming, stress memory, and their regulation by various epigenetic mechanisms, including DNA methylation, chromatin modifications, non-coding RNAs, and RdDM machinery. Until now, the exact mechanism of the establishment and maintenance of stress memory has been unclear. I attempted to synthesize knowledge in the field, pointing out that the central role likely belongs to RdDM machinery and nucleosome occupancy, regulated in a positive and negative manner by multiple proteins.

Many questions still remain. Is the memory of stress gradual (linear) or incremental with thresholds? How do exposed plants perceive the stress signals, and how do they measure the length of treatment? How is the somatic reprogramming prevented when stress is no longer present? How do histone modifying enzymes promote changes in a locus-specific manner?

I tried to be thorough in my literature analysis, but I have likely missed many worthy papers; thus, I apologize to the scientists whose work I was not able to cite in this review.

## Figures and Tables

**Figure 1 plants-12-03667-f001:**
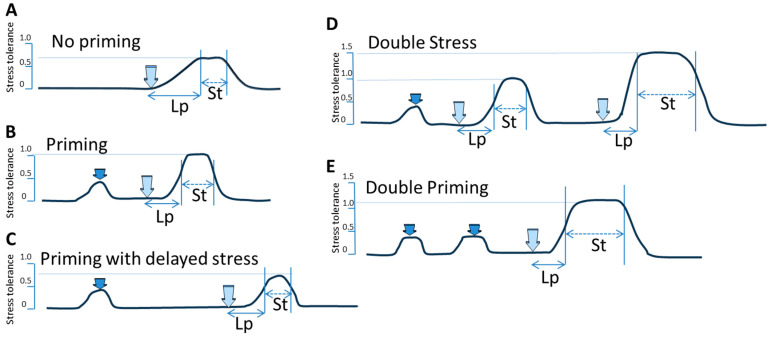
Schematic presentation of stress tolerance with and without priming. A light blue long arrow pointing down indicates the stress exposure. Short dark blue arrow shows priming. “Lp”—latent period (shown as solid double-ended arrow), time required to mount stress tolerance; “St”—stress tolerance (shown as dashed double-ended arrow). Y axis shows arbitrary scale (from 0 to 1.5) indicating the level of stress tolerance. (**A**) “No priming” is characterized by long Lp and relatively low (at arbitrary level of ~0.7) and short-lasting St. (**B**) “Priming” is characterized by shorter Lp and higher (at arbitrary level of ~1.0) and longer lasting St. (**C**) “Priming with delayed stress”—likely similar in effect to “No priming” or somewhere between “Priming” and “No priming”. (**D**) “Double stress”—first stress response after priming is characterized by shorter Lp and stronger, longer St, similar to “Priming”, but the second stress exposure results in even stronger (arbitrary level of ~1.5) and longer lasting St. (**E**) “Double Priming” is characterized by shorter Lp and stronger (arbitrary level of ~1.2) and longer lasting St, which is somewhere between “Priming” and “Double stress”.

**Figure 2 plants-12-03667-f002:**
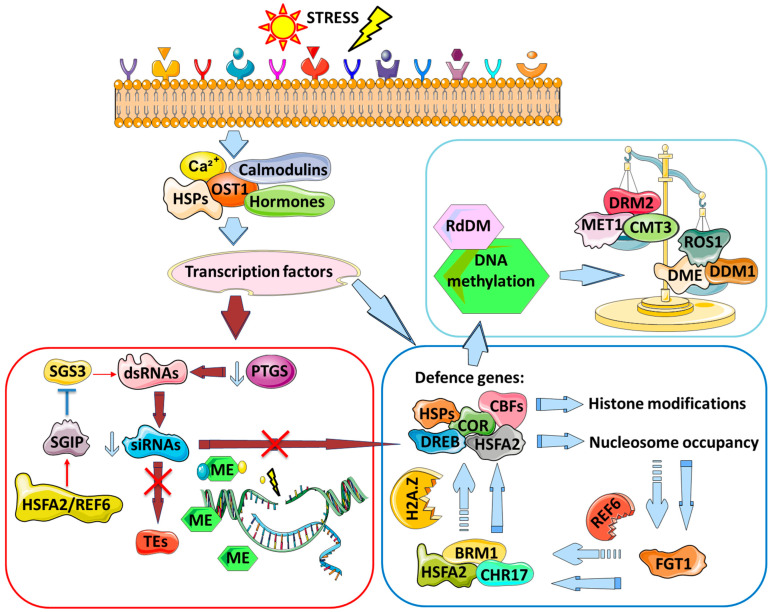
Potential mechanisms of the response to stress and activation of stress memory. Stress is perceived at many levels, including changes in Ca^2+^ in the cell, the activation of calmoldulins, changes in protein folding, hormonal levels, and many more. This results in the activation of various stress-specific transcription factors that on one side activate stress-responsive genes, and on another side, release the RdDM regulation of transposons and various stress-responsive genes. The activation of HSPs/COR/CBFs/DREB and other proteins results in changes in histone modifications and nucleosome occupancy, which, in turn, activates proteins like FGT1 and REF6 that activate proteins like BMR1, CHR17, and HSFA2 that are able to maintain pockets of permissive chromatin and low nucleosome occupancy, leading to stress memory. A decrease in methylation at transposons results in an increase in the production of dsRNAs that would typically be processed to siRNAs. Heat stress activates HSFA2 that results in the inhibition of the SGS3 protein involved in siRNA production, resulting in a decrease in the number of produced siRNAs and, thus, a decrease in methylation and the suppression of transposons and stress-responsive loci (indicated by crossed red arrows), allowing the stress memory to persist. Finally, transcription factors activate RdDM and DNA methylation machinery that establish a differential pattern of DNA methylation/gene expression at stress-responsive loci. The maintenance of the memory of stress would depend on the equilibrium of DNA methyltransferases, enzymes removing methylation marks such as ROS1 and DEMETER (DME) [46], and chromatin modifiers like DDM1.

**Table 2 plants-12-03667-t002:** Factors involved in the establishment, maintenance, and eraser of the somatic stress memory.

Mechanisms	Proteins	Promoter/Establisher	Blocker/Eraser
**Chromatin/nucleosome**	BRM/CHR17	X	
	BRU1	X	
	FGT1	X	
	HFSA2	X	
	H2A.Z	X	
	CAF-1		X
	MOM1		X
**RdDM**	SGS3		X
	DRD1/polIVb	X	X
**DNA methylation**	DDM1		X
	ROS1	X	X
**Histones**	H3K4me3	X	X
	H3K18ac	X	X
	H3K27me3		X
	H3K9me3		X
	H3K36me3		X
	H2A.Z	X	

## Data Availability

Not applicable.

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
