# Peer review of "Role of Epigenetic Factors in Response to Stress and Establishment of Somatic Memory of Stress Exposure in Plants"

_plants, 2023, doi:10.3390/plants12213667_

Round 1
Reviewer 1 Report
In this manuscript, Prof Kovalchuk give an very nice overview of how key epigenetic factors are involved in establishing a stress response and memory in plants. In general, the manuscript is very well written, easy to follow and it nicely covers different aspect of plant stress response, including priming. I do not have any major concerns but only few minor comments/questions.
- Line 118 - 128: concerning “forgetting” of a stress exposure I wondering whether it cannot be simply related to a progressive dilution of particular epigenetic marks following successive cell division ?
- Line 152 – 153: “the change of affinity of histone to DNA” is only true for acetylation (i.e., it reduces positive electric charges on histone thus increasing repulsive forces between histone and DNA) since methylation do not directly affect nucleosome physical properties but needs “readers” to recruit chromatin remodeling factors.
- Line 165 – 166: I would suggest to be less restrictive concerning conclusions from the work by Pecinka et al., 2010 since only 2 histone modifications (H3K9me2 and H3K4me3) were tested.
- - Line 187 – 189: Related to the question of the heterochromatin decondensation I personnaly found very interesting the work by Tessadori et al., 2009 (DOI: 10.1371/journal.pgen.1000638) in which authors found that Arabidopsis accessions from low latitude (high irradiation/light intensity) present low chromocenter compaction.
- Line 241: please correct “draught”.
- Finally, related to BRM and (heat) stress memory, I found very interesting the recent connection between the protein kinase TOR, stress response and epigenetic processes proposed in 2 related articles. In Sharma et al., 2019, TOR was found to promote the long-term accumulation of H3K4me3 at thermomemory-heat stress genes (DOI: 10.1104/pp.18.01371). In Dong et al., 2023, TOR was reported to modulate CLF and BRM activity to regulate the expression of stress-related genes (DOI: 10.1093/jxb/erac486).
Author Response
- Line 118 - 128: concerning “forgetting” of a stress exposure I wondering whether it cannot be simply related to a progressive dilution of particular epigenetic marks following successive cell division ?
Response:
Very good point; I have introduced it into the text.
- Line 152 – 153: “the change of affinity of histone to DNA” is only true for acetylation (i.e., it reduces positive electric charges on histone thus increasing repulsive forces between histone and DNA) since methylation do not directly affect nucleosome physical properties but needs “readers” to recruit chromatin remodeling factors.
Response:
You are right. We have changed the text to be more precise. Now it reads: Various modifications of histone tails either result in the change of affinity of histones to DNA (histone acetylation) or change the tail structure allowing to recruit different kind of readers (histone methylation), thus regulating chromatin condensation and gene expression.
- Line 165 – 166: I would suggest to be less restrictive concerning conclusions from the work by Pecinka et al., 2010 since only 2 histone modifications (H3K9me2 and H3K4me3) were tested.
Response:
We are not sure we understand the concern of the reviewer. We wrote that activation of repetitive elements occurred without changes in DNA methylation and minor changes in histone modifications, thus suggesting it has likely occurred due to nucleosome repositioning.
- - Line 187 – 189: Related to the question of the heterochromatin decondensation I personnaly found very interesting the work by Tessadori et al., 2009 (DOI: 10.1371/journal.pgen.1000638) in which authors found that Arabidopsis accessions from low latitude (high irradiation/light intensity) present low chromocenter compaction.
Response:
Thank you for the reference. We agree, a great paper; we wrote the following in the new version: “The important role of differential chromatin compaction for the response to stress was demonstrated in the work of Tessadori et al. (2009). The authors analyzed 21 accessions of Arabidopsis grown on different latitude (and therefore, different light intensity) and found direct positive correlation between latitude and the level of chromatin condensation; the higher was the latitude, the higher was chromocenter compaction, and vice versa [20]. The authors suggested that such changes in chromatin compaction is part of the acclimation process in Arabidopsis.”
- Line 241: please correct “draught”.
Response:
Oups, thanks for catching this typo.
- Finally, related to BRM and (heat) stress memory, I found very interesting the recent connection between the protein kinase TOR, stress response and epigenetic processes proposed in 2 related articles. In Sharma et al., 2019, TOR was found to promote the long-term accumulation of H3K4me3 at thermomemory-heat stress genes (DOI: 10.1104/pp.18.01371). In Dong et al., 2023, TOR was reported to modulate CLF and BRM activity to regulate the expression of stress-related genes (DOI: 10.1093/jxb/erac486).
Response:
Thank you for the references. We introduced both of them.
“There is also a link between energy sensor TARGET OF RAPAMYCIN (TOR) and BRM. TOR appears to repress genes associated with bistable chromatin state, driven by H3K4me3 and H3K27me3, and thus actively regulates transcriptional response to stress. BRM, together with CURLEY LEAF (CLF), which deposits H3K27me3 marks, activate TOR-repressed genes, allowing rapid response to stress [31].”
“Another link between activation of heat shock proteins and heat-stress memory appears to be the glucose and the energy sensor TOR-E2Fa signalling module [48]. In response to heat, glucose can promote chromatin acetylation of H3K4 trimethylation. Glucose-primed accumulation of H3K4me3 at heat stress memory-associated loci is mediated through HLP1 and requires the activity of TOR [48].”
Reviewer 2 Report
1) The review presents a general and broad description of the role of epigenetic factors in plant stress response without offering novel insights or a unique perspective. It doesn't provide a clear indication of the specific focus or novel contribution of the paper.
2) If only there is one author, there should not be we in lines L53 and L100.
3) Page 2, Line58: What about chloroplasts?
4) P2, L63: By definition, plants are not sedentary.
5) P3, L78: The reference cited is not the most appropriated to that process
6) P3, L88: The reference cited is not the most appropriated to that process
7) All the introduction´ citations are from review of the topics.
8) The review lacks depth and specific details about the mechanisms involved in epigenetic responses to stress. It briefly mentions mechanisms like non-coding RNAs, DNA methylation, histone modifications, and nucleosome repositioning, but it doesn't delve into these mechanisms in detail or propose new findings.
9) The author utilizes 'adaptation' and 'acclimatization' interchangeably, but it's essential to distinguish between these terms in the context of plants, as they represent distinct processes.
10) The review could benefit from a more comprehensive coverage of relevant literature to showcase a deep understanding of the topic. This would involve referencing recent studies and advancements in the field to provide a well-rounded foundation for the discussion. Examples of this is the missing references in all the manuscript, just to mentions few examples in lines (L): L26, L29, L32, L36, L59, L67, L74, L118, L123…..
11) The review lacks a clear and organized structure. It should present information in a more coherent manner, making it easier for readers to follow the logical flow of the paper and understand the key points.
Sentences too long in some places.
Author Response
1) The review presents a general and broad description of the role of epigenetic factors in plant stress response without offering novel insights or a unique perspective. It doesn't provide a clear indication of the specific focus or novel contribution of the paper.
Response
Thank you for your constructive criticism. The novelty of this review is in combination of the analysis of the mechanisms of priming and epigenetic regulation of response to stress. We also summarized various factors and their role in somatic memory and suggested several potential mechanisms of its maintenance as well as eraser.
2) If only there is one author, there should not be we in lines L53 and L100.
Response
I fixed it everywhere.
3) Page 2, Line58: What about chloroplasts?
Response
Of course. This is now added.
4) P2, L63: By definition, plants are not sedentary.
Response
This is semantics of course, but if the reviewer does not like the definition, we can remove it. We changed it to: “Avoidance is not part of plant’s mechanism of response to stress”.
5) P3, L78: The reference cited is not the most appropriated to that process
Response
We cited a review covering metabolic response to stress, but we of course can be more specific. We now cite the following paper:
Xu Y, Freund DM, Hegeman AD et al (2022) Metabolic signatures of Arabidopsis thaliana abiotic stress responses elucidate patterns in stress priming, acclimation, and recovery. Stress Biol 3:86–102
6) P3, L88: The reference cited is not the most appropriated to that process
Response
We only partially agree, but we have modified the text slightly and added additional references to be more broad (not only about extremophiles).
7) All the introduction´ citations are from review of the topics.
Response
Introduction only has one reference. The reviewer probably refers to the section “2. General stress response; priming”? In this section, several references were not the reviews. We added additional “original research” articles to this section.
8) The review lacks depth and specific details about the mechanisms involved in epigenetic responses to stress. It briefly mentions mechanisms like non-coding RNAs, DNA methylation, histone modifications, and nucleosome repositioning, but it doesn't delve into these mechanisms in detail or propose new findings.
Response
Hmm, it is hard to agree. We provided information on chromatin condensation, nucleosome repositioning, role of CRFs, the role of chromatin modifiers lacking ATPase domain, role of histone modifications, role of DNA methylation and role of RdDM/siRNAs with many specific details, and summarised in tables and a figure. We of course could go into the details on each mechanism, but this review would be much much larger – it is already over 12,000 words. Still, we added information in several places in the manuscript.
9) The author utilizes 'adaptation' and 'acclimatization' interchangeably, but it's essential to distinguish between these terms in the context of plants, as they represent distinct processes.
Response
Very important point, and I apologize for being sloppy about it. I have clarified this in the text and added the definitions into the Box 1.
10) The review could benefit from a more comprehensive coverage of relevant literature to showcase a deep understanding of the topic. This would involve referencing recent studies and advancements in the field to provide a well-rounded foundation for the discussion. Examples of this is the missing references in all the manuscript, just to mentions few examples in lines (L): L26, L29, L32, L36, L59, L67, L74, L118, L123…..
Response
Some of these sentences are common knowledge, but we added references and information to all suggested places.
11) The review lacks a clear and organized structure. It should present information in a more coherent manner, making it easier for readers to follow the logical flow of the paper and understand the key points.
Response
This is hard to address, but I made an effort to streamline the review. I started with general intro and continued with general response to stress and role of priming. I then introduced somatic memory of stress and suggested that I will cover epigenetic mechanisms. I started with chromatin condensation/nucleosome occupancy, followed by the role of chromatin modifiers, histone modifications, DNA methylation and the role of RdDM/ncRNAs. In the revision, I introduced potential role of methylated RNA. I have concluded the review by summarising the information about potential inducers/blockers/erasers of the somatic memory and presented some proposed mechanisms in Figure 2. I believe this flow is clear and structured.
Reviewer 3 Report
In present article, the author have reviewed the role of epigenetic factors in response to stress tolerance, priming role and somatic memory development in response to various stress incidences. As a living organisms modify their fundamental DNA code and protein using various chemical functional groups. This helps to survive and adapt to the local growth conditions.
The review is well constructed and organized, however it needs some changes.
Some comments:
1. In abstract author should mention the types of epigenetic modifications and correlate it to the priming effects.
2. In Figure 1, the abbreviation ‘lp’ could be capitalized.
3. It would be good, if author could provide statistical information epigenetic modification and its impact.
4. Does priming or stress affect RNA methylation? Author should clarify this. Refer to article PMID: 36938064.
5. Author could include additional figures, explaining cellular mechanisms of cellular epigenetics changes in protein, DNA and RNA molecules.
6. Figure 2 is too vague/too simple, it should be refined.
Minor language editing is suggested.
Author Response
Some comments:
- In abstract author should mention the types of epigenetic modifications and correlate it to the priming effects.
Response
We added this information.
- In Figure 1, the abbreviation ‘lp’ could be capitalized.
Response
Done
- It would be good, if author could provide statistical information epigenetic modification and its impact.
Response
Your suggestion is very relevant and interesting. It would be wonderful to have it, but I am afraid it is not possible. We looked through many publications, and there is very little to no information on the percentages of DNA methylation or histone mods at specific loci and their impact on function, gene expression or phenotype. I think we are too early in our understanding how individual epigenetic events affect function. I guess some of it can be done through in-depth bio-informatic/statistical analysis, but would require considerable time to complete.
At the same time, our Table 1 and 2 provide some (but not quantifiable) information.
- Does priming or stress affect RNA methylation? Author should clarify this. Refer to article PMID: 36938064.
Response
This is an excellent suggestion. We added an entire section in the text:
3.5 Potential role of methylated RNA in stress response
RNA methylation is a common process, well-described in animals, but still poorly understood in plants [131]. Methylated adenine is most common (m6A), but cytosines (m3C) can also be methylated. Plants encode all classcial writers, erasers and readers of methylation. The importane of these proteins is shown by various phenotypic abnormalities, including altered growth [132, 133] and response to pathogens [134].
Thousands of genes were reported to be affected by RNA methylation [135]; this is especially common in organelles, almost all chloroplast transcirpts and 86−90% mitochondrial transcripts are methylated, suggesting a critical role of this process in the energy metabolism [136]. Methylation of ribosomal RNA in chloroplasts is crucial for chloroplast biogenesis, photosynthesis and for the proper abcisic acid response [137].
Methylation of mRNA may alter its mobility. m5C-modified TCTP1 (TRANSLATIONALLY CONTROLLED TUMOR PROTEIN 1) was shown to be actively transported to roots and change root growth [138]. Methylated mRNAs can move long distances and thus could serve as developmental and stress response signals. In Arabidopsis, mutation of m6A writers resulted in sensitivity to salt stress that correlated with the level of m6A [139]. One of the writers, VIR protein, modulated the balance of reactive oxygen species by destabilizing the mRNA negative regulators of salt stress such as ATAF1, GI and GSTU17 [139]. Similarly, exposure of barley roots to cadmium stress resulted in significant changes in m6A levels in over 400 genes, including Cd-responding regulatory genes of MAPK, WRKY and MYB groups [140].
Although there are not many reports indicating the direct role of methylated RNA in response to stress and stress memory, m6A and m3C in stress responsive mRNAs and perhaps miRNAs [141] have to be considered as potential candidates.
- Author could include additional figures, explaining cellular mechanisms of cellular epigenetics changes in protein, DNA and RNA molecules.
Response
I feel that presenting basic information on type of epigenetic changes and the mechanism of their occurrence is beyond the scope of this review.
- Figure 2 is too vague/too simple, it should be refined.
Response
I have modified the figure, adding more information. At the same time, the process of acquiring somatic memory is complex and likely stress-specific. So, there is no way we can comprehensively cover the mechanisms in one figure.
Round 2
Reviewer 2 Report
I have carefully reviewed the manuscript and while the content is promising, I must emphasize the need for significant improvement in the quality and design of the figures. The current figures are of poor quality and lack clarity, making it difficult for readers to comprehend the presented data effectively.
Author Response
We modified both figures (Figure 2 more substantially) as well as provided additional explanations to both figures. We hope it is much better now.